# Silencing of dentate gyrus inhibits mossy fiber sprouting and prevents epileptogenesis through NDR2 kinase in pentylenetetrazole kindling rat model of TLE

Chen Zhang[1], Zixian He[2], Zheren Tan[3]☯*, Fafa Tian[1]☯*

1 Departments of Neurology, Xiangya Hospital, Central South University, Changsha, China, 2 Biomedical Engineering major, College of Engineering, Boston University, Boston, Massachusetts, United States of America, 3 Departments of Critical Care Medicine, Xiangya Hospital, Central South University, Changsha, China

☯ These authors contributed equally to this work.
* xysntff@sina.com (FT); tanzheren@126.com (ZT)

**Data Availability Statement:** All relevant data are within the manuscript and its Supporting information files.

## Abstract

Epileptogenesis is a potential process. Mossy fiber sprouting (MFS) contributes to epileptogenesis. Silencing of the dentate gyrus (DG) suppressed spontaneous seizures model of epilepsy and hyperactivity of granule cells resulted in MFS in vitro. However, the role of DG's excitability in epileptogenesis have not yet been well explored, and underlying mechanisms has not been elucidated. Using chemical genetics, we studied whether MFS and epileptogenesis could be modulated by silencing of DG in the PTZ kindling rat model of epilepsy. MFS and protein expression was measured by Timm staining, Western blotting, and Immunofluorescence. Previous studies demonstrated that MFS and epileptogenesis could be modulated by a regulator of axonal growth (e.g. RGMa, PTEN). NDR2 kinase regulate neuronal polarity and prevents the formation of supernumerary axons in the hippocampus. We experimentally confirmed chemogenetic inhibition in DG resulted in decreased MFS and NDR2 expression, and alleviated epileptogenesis. Furthermore, our results showed that injection of AVV vector expressing NDR2 into DG induced upregulation of NDR2 in the hippocampus, and over expression of NDR2 in the hippocampus promote MFS and block protective effect of chemogenetic silencing of DG on epileptogenesis. Overall, we concluded that silencing of DG inhibits MFS and prevents epileptogenesis through NDR2 in the hippocampus in the PTZ kindling rat model of TLE, thereby providing a possible strategy to prevent epileptogenesis.

## 1. Introduction

Epilepsy is a devastating neurological disorder that affects more than 60 million people worldwide. Despite the introduction of many new antiepileptic drugs in the last years, approximately 20–30% of epileptic patients fail to achieve seizure free even with optimal medical treatment, finally developing refractory epilepsy [1, 2]. The commonest type of refractory epilepsy that

**Funding:** This study was supported by National key R&D program of China (2017YFC1310003) and Natural Science Foundation of China (81571276). The funders had no role in study design, data collection and analysis, decision to publish, or preparation of the manuscript.

**Competing interests:** The authors have declared that no competing interests exist.

can be found in adult patients is temporal lobe epilepsy (TLE) [3]. However, the precise mechanisms of epileptogenesis of TLE are still unknown. In this regard, there is an urgent need to investigate the epileptogenesis of TLE for identifying more specific and promising therapeutic targets for epilepsy.

Mossy fiber sprouting (MFS) is one of the common pathogenic manifestations in the dentate gyrus of both patients and animal models with TLE [4]. Previous studies had demonstrated the aberrant axonal outgrowth caused by MFS is the cause of the imbalance between excitatory and inhibitory inputs in the hippocampus, which contributes to the epileptogenesis of TLE [5–7]. It has been demonstrated that axonal outgrowth was modulated by Repulsive guidance molecule a (RGMa) and phosphatase and tensin homolog (PTEN) [8, 9], MFS and epileptogenesis of TLE were suppressed or promoted by up-regulation of RGMa [10] and down-regulation of PTEN [11] respectively. Those findings implicated that regulators of axonal growth may be a potential target to prevent and cure TLE.

The dentate gyrus (DG) plays an important role in epilepsy because of its function to protect hippocampal circuits from over excitation. Besides, optogenetic inhibition of granule cells is sufficient to inhibit spontaneous seizures in a mouse model of TLE [12]. However, the role of DG's excitability in epileptogenesis has not yet been well explored. Aberrant axonal outgrowth of granule cells in DG caused MFS. Previous studies had confirmed that axonal morphological plasticity was regulated by the excitability of neurons [13], and hyperactivity of granule cells resulted in the sprouting of mossy fiber in vitro [14]. Therefore, we hypothesized that silencing granule cells in DG relieve or prevent the epileptogenesis of TLE, and MFS may be involved in it.

On the other hand, Nuclear Dbf2-related 2 kinase (NDR2, also known asSTK38L) are novel members of the core cassette of Hippo signaling. NDR2 could be activated by interaction with Mps One binder 1 (MOB1) [15], which was a regulator of neurite outgrowth [16], and Yang et al. found that NDR kinases regulate neuronal polarity and prevent the formation of supernumerary axons in the hippocampus [17]. By considering the confirmed function of NDR2 in axonal outgrowth, we conjectured that NDR2 acts as a key molecules in MFS and epileptogenesis of TLE.

In summary, we supposed that silencing granule cells in DG relieve MFS and prevent the epileptogenesis of TLE through NDR2. Chemogenetics involve the selective expression of modified receptors, such as Designer Receptors Exclusively Activated by Designer Drugs (DREADD), in specific cell types and regions of the brain, allowing selective modulation of neurons using administration of clozapine N-oxide (CNO). We tested those hypotheses by silencing neurons in DG and regulating the expression of NDR2 in pentylenetetrazole (PTZ) kindling rat model of TLE.

## 2. Materials and method

### 2.1 Ethic statement

All animals were treated humanely. Study design and all animal experimental protocols were approved by the Ethics Committee of Xiangya Hospital, Central South University, Changsha, China(Protocol Number: 20153213), which are in accordance with the guidelines of the National Institutes of Health for the care and use of laboratory animals, based on the guidelines of the World Medical Association Declaration of Helsinki. All surgery was performed under anesthesia with isoflurane, and all efforts were made to minimize suffering. Humane endpoints were: weight loss of more than 20%, no food intake, and weakening of reflexs. The rats were monitored daily for these humane endpoint. If the rat meets the criteria, it will be euthanized within 10 minutes. Training in animal care and observation was provided by the Department of experimental animals, Central South University.

## 2.2 Animals

Male wild-type Sprague–Dawley rats (2–4 months old) were housed in the Center for Experimental Animals of Central South University (Changsha, China) under controlled conditions (18–25˚C; 50–60% humidity; 12 h light/dark cycle) with food pellets and water available ad libitum. Treatment and care complied with the ARRIVE (Animal Research: Reporting of In Vivo Experiments) guidelines.

In this study, we randomly divided adult rats into four groups: I) DREADD+Vehicle group (one of the control groups), in which rats were pre-treated with AAV-hM4Di vector in DG and then administered PTZ(35 mg/kg, ip) and vehicle (ip) 4 weeks later; II) the Non-DREADD+CNO group(one of the control groups), in which rats were pre-treated with control vector in DG and then administered PTZ (35 mg/kg, ip) and CNO(10 mg/kg, ip) 4 weeks later; III) DREADD+CNO group, in which rats were pre-treated with AAV-hM4Di vector in DG and then administered PTZ (35 mg/kg, ip) and multiple doses of CNO(1,5,10 mg/kg, ip) 4 weeks later; IV) DREADD+CNO+NDR2 group, in which rats were pre-treated with AAV-hM4Di vector and AVV vector expressing NDR2 in DG and then administered PTZ (35 mg/kg, ip) and CNO(10mg/kg, ip) 4 weeks later. 109 rats were used in the present study, of which 13 died of severe seizures, and the others were euthanized after the last observation of PTZ-induced seizure.

## 2.3 Establishment of PTZ kindling rat model

The establishment of the PTZ kindling rat model was performed as described in our previous study [18]. Briefly, a subconvulsive dose (35 mg/kg, i.p.) of PTZ (Sigma-Aldrich, USA) was administered repeatedly to rats in each group every day until kindling or sacrifice occurred. Seizure stage classification was based on a modified description of the Racine scale: 0-no responses, 1-mouth and facial cramps, 2-head nodding, 3-forelimb clonus, 4-rearing, and 5-rearing and falling [19]. A score $\geq 3$ for 5 consecutive days indicated that the rat was kindled. All rats were sacrificed 4weeks after the start of the model establishing for timm staining and immunohistochemistry analysis. Severe seizure latency was measured as the time between the first injection of PTZ and the onset of the severe seizure which was defined as seizures with a Racine scale of 3 or higher.

## 2.4 Virus delivery

Rats were transported to the operating room and anaesthetized by using isoflurane and then placed in a stereotaxic device (Yuyan Instruments, China). For the pharmaco-genetic inactivation of neurons, the DREADD rats were stereotactically microinjected with 0.5μl AAV-hM4Di vector (pAVV-hSyn-HA-hM4Di-mCherry; 3.18E+12 genome-copies/ml; provided by OBiO Technology, China) into DG (from bregma, antero-posterior: -2.9mm, medial lateral: ±0.8mm, ventral from brain surface: -4.6mm) during surgery. pAVV-hSyn-HA-mCherry (0.5μl; 1.91E+13 genome-copies/ml: provided by OBiO Technology, China) was injected in DG as a control vector in Non-DREADD rats.

To up-regulate the expression of NDR2, rats in the DREADD+CNO+NDR2 group were stereotactically microinjected with a mixture of 0.5μl AAV-hM4Di vector and 0.5μl AVV vector expressing NDR2 (pAVV-hSyn-EGFP-P2A-stk38l-3Xflag-WPRE; 1.0E+12 genome-copies/ml; provided by OBiO Technology, China) into DG.

The vectors contained a hSyn promoter to guide specific expression in neurons of the DG.

The above viral delivery was injected via an injection pump with a 1μL syringe at 100 nl/min. After each injection, the needle was left in place for 5 min to prevent backflow of the virus and then slowly withdrawn. At the end of the injection, the skin was sutured and then

sterilized with an iodine solution. After injection, the rat was transferred back to its home cage for recovery, and establishment of the PTZ kindling model was performed 4 weeks after surgery.

## 2.5 Pharmacology

PTZ and CNO were purchased from Sigma-Aldrich (St. Louis, MO, USA). PTZ was dissolved in saline, and CNO was dissolved in 5% DMSO solution. PTZ was administrated intraperitoneally (i.p.) at 35 mg/kg and CNO was were administrated intraperitoneally (i.p.) at 1,5,10 mg/kg after daily behavioral seizure observation during PTZ kindling.

## 2.6 Behavioral seizure observation

Seizure responses were observed over 30 minutes after PTZ injection and classified according to the modified Racine score. Rats that died during PTZ kindling were excluded from the comparative analysis.

## 2.7 Tissue processing

Tissue processing, Immunofluorescence, Western blotting and Timm staining were performed as described in our previous studies [18, 20]. Briefly, rats were deeply anesthetized by intraperitoneal injection of ketamine/xylazine (100/10 mg/kg). Rats were perfused intracardially with 250 ml saline for immunohistochemistry or with 250 ml saline and 250 ml 0.4% sodium sulfide in 0.1 M phosphate buffer (pH 7.3) for Timm staining, then followed by application of 250ml 4% paraformaldehyde for perfusion fixation. The brains were removed, fixed in 4% paraformaldehyde for 24 h, dehydrated in 30% sucrose, and sectioned into 20μm coronal sections (frozen cryosections). For western blotting, Hippocampus was separated and was stored at -80˚C

## 2.8 Immunofluorescence

Slices were removed from storage at -80˚C and dried in air. Slides were washed 3 times in PBS for 5 min. The slides were blocked with 5% donkey serum in PBST (PBS Triton-X 100) and incubated at room temperature for 2 h, then incubated overnight with the following primary antibodies at 4˚C: rabbit anti-NDR2 (1:500, Affinity, USA). Next, at room temperature, the slices were incubated with the following secondary antibodies for 2 h: goat antirabbit secondary antibodies (1:500, MultiScience, China). Then, nuclei were stained with Hoechst 33342 for 5 min. Finally, the sections were mounted with 50% glycerin. Three slices were selected randomly from each rat. Photomicrographs were captured for each animal with a microscope. The mean optical density values of the areas with positive immunolabeling were measured using ImageJ software.

## 2.9 Western blotting

The proteins were extracted from hippocampal tissues using RIPA lysate buffer (Beyotime, China), and then concentrations of proteins were measured using bicinchoninic acid (BCA) Protein Assay Kit (Beyotime, China). Proteins were separated by using sodium dodecyl sulfate polyacrylamide gel electrophoresis (SDS-PAGE) and then transferred to nitrocellulose membranes, which had been washed by using Tris Buffered Saline Tween (TBST), and then were blocked with 5% skimmed milk in TBST (room temperature, 2 h), The membranes were incubated at 4˚C overnight with primary antibodies. The primary antibodies included rabbit anti-NDR2 (1:3000, Affinity, USA), and rabbit anti-GAPDH (1:10,00, Servicebio, China). Unbound

antibodies were washed by TBST, then the membranes were subsequently incubated with HRP-labeled goat antirabbit IgG secondary antibodies (1:200, Beyotime, China) for 1 h. The immunoreactive bands were visualized by using a chemiluminescence (ECL) kit (Beyotime, China), Images were acquired with a Chemiluminescent Gel Imaging System (FluorChem FC3, ProteinSimple, USA), and densitometric analysis was performed with ImageJ software.

### 2.10 Timm staining

In a darkroom, corresponding sections were stained for 45 min in a specific solution which was composed of 100 ml 50% gum arabic, 5 ml citrate buffer (27.2% citric acid and 31% sodium citrate), 15 ml 5.6% hydroquinone, and 0.5 ml 17% silver nitrate. After washing in water, sections were routinely dehydrated, cleaned and mounted with gum. Finally, the corresponding Timm's score for the CA3 region and DG in the hippocampus was analyzed based on the published criteria [21].

### 2.11 Statistical analysis

Data are expressed as the mean and standard deviation or the median and interquartile range. The Shapiro-Wilk test and Kolmogorov Smirnov test were used to evaluate the normal distribution of the data, and the Levene test was used to compare variances before we used a parametric test to compare the differences between groups. If data did not fulfill the requirements of a parametric test, a nonparametric test would be selected. We used the independent sample t test or Mann-Whitney test to determine the differences in the groups. kaplan-meier survival was used to determine the difference in kindling tendency. The Kruskal-Wallis test followed by Bonferroni Procedure was used to account for multiple comparisons. The results were considered statistically significant when $P < 0.05$. All statistical analyses were two-sided and were conducted using the Statistical Package for the Social Sciences version 23.0. Graph-Pad Prism 8 was used to make graphs.

## 3. Results

### 3.1 Histological verification of virus infection in the hippocampus

We first verified the expression of the DREADD-associated virus in the hippocampus. As shown in Fig 1, our virus injections resulted in predominant mCherry fluorescence within the DG.

### 3.2 The effect of chemogenetic inhibition in DG in PTZ kindling rat model

We conducted comparative experiments in both, rats injected with hM4Di vector as well as rats, injected with control vector. To determine if the chemogenetic approach can affect epileptogenesis in PTZ kindling rat model, mortality, kindling rate, latency and tendency of kindling, and seizure behaviors (stage of seizures, severe seizure latency) were compared between control and experimental groups.

Administration of CNO (10 mg/kg) in the DREADD+CNO group(n = 22) does not affect mortality and severe seizure latency compared with the Non-DREADD+CNO group(n = 12) and DREADD+Vehicle groups (n = 11, Fig 2A and 2D). A trend toward a lower kindling rate (form 91.7% to 63.6%, $p = 0.059$) was observed in the DREADD+CNO group (CNO at 10 mg/kg) as compared with the DREADD+Vehicle group (Fig 2B). Longer latency and smoother tendency of kindling were observed in the DREADD+CNO group compared with DREADD+Vehicle and Non-DREADD+CNO groups, respectively (Fig 2C and 2E). Besides, we observed a significant decrease in seizure stage at days 6,7,8,10,17, and days 5,6,7,8 in the DREADD+CNO group when compared to DREADD+Vehicle and Non-DREADD+CNO

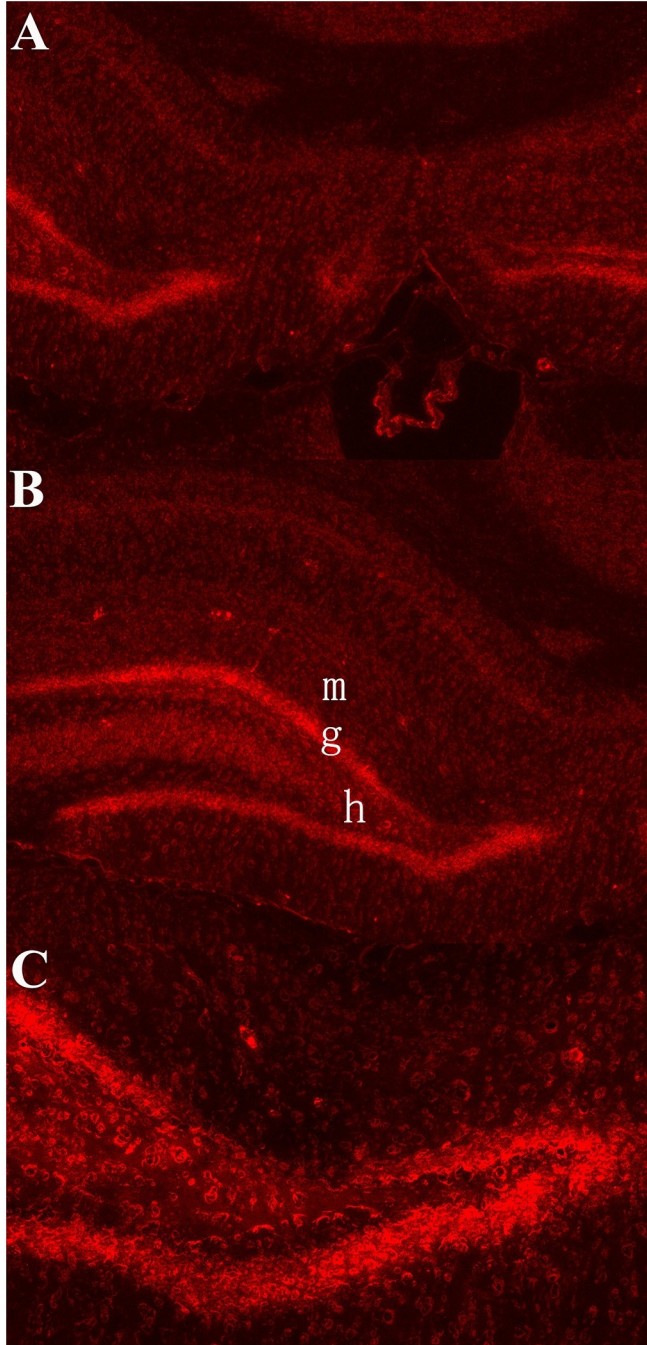

**Fig 1. Histological verification of virus infection in hippocampus.** (A and B) Representative photomicrographs showing mCherry fluorescence within the hippocampus. (C) Higher magnification images showing mCherry fluorescence in the DG. m, molecular layer; g, granule cell layer; h, hilus; Red = mCherry; DG, dentate gyrus.

groups, respectively (Fig 2F). Those findings indicated that the chemogenetic inhibition in DG alleviates epileptogenesis in the PTZ kindling rat model.

The MFS and NDR2 expression in the hippocampus was investigated. MFS in CA3 was significantly down-regulated in DREADD+CNO group(n = 6) compared with that in control group(n = 5) (median Timm score: 2.0 vs 1.0, Fig 3A and 3B), and no difference in MFS was

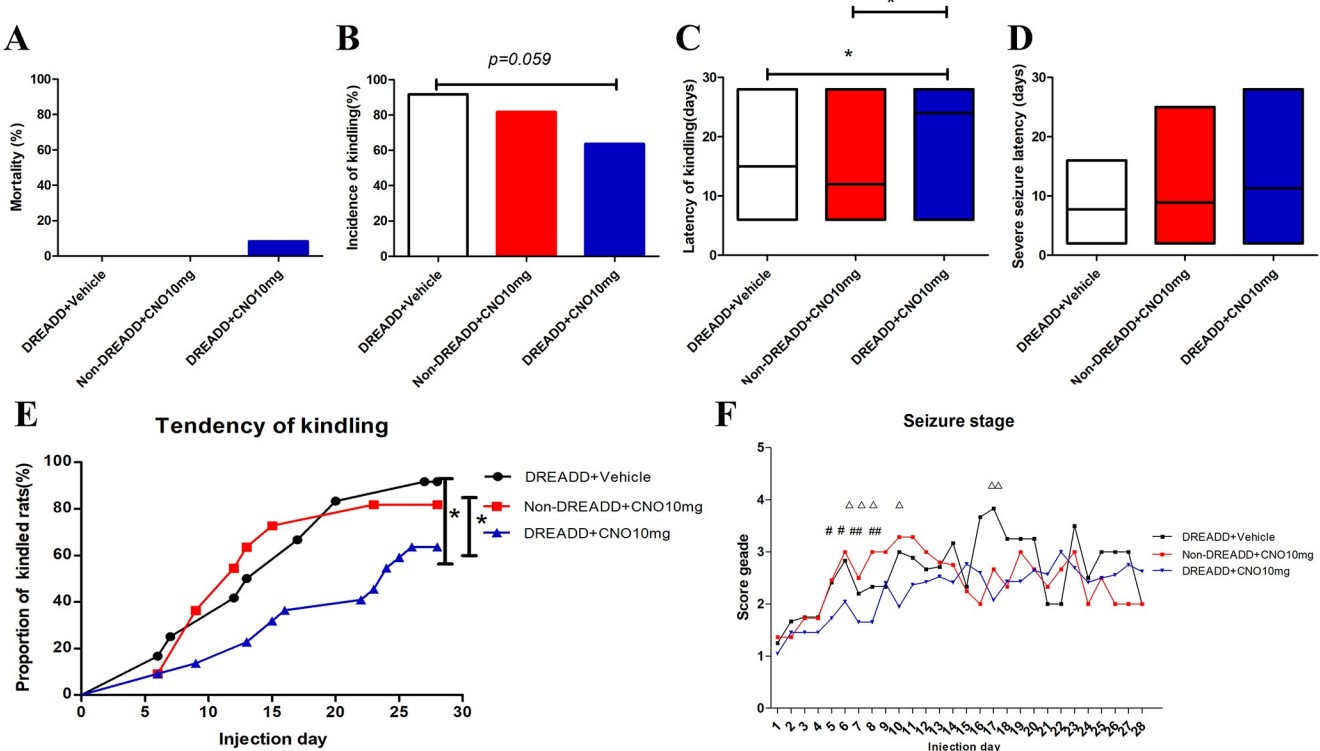

**Fig 2. CNO administration alleviates epileptogenesis in PTZ kindling rat model infected with hM4Di vector in DG.** Mortality(A), kindling rate(B), latency of kindling(C), severe seizure latency(D), tendency of kindling(E) and seizure stage(F) of two control groups and DREADD+CNO group. *indicates $P < 0.05$, **indicates $P < 0.01$;ΔIndicates $P < 0.05$, ΔΔindicates $P < 0.01$,DREADD+CNO group vs DREADD+Vehicle group;# Indicates $P < 0.05$, DREADD+CNO group vs Non-DREADD+CNO group; CNO, clozapine N-oxide.

observed in DG between the two groups. Data showed that the hippocampal level of NDR2 was significantly decreased in the DREADD+CNO group(n = 4) compared with that of the control group (n = 4, median relative level:0.90 vs 0.98, Fig 4A). In addition, immunofluorescence analysis demonstrated that the level of NDR2 in the DG and CA3 of the hippocampus was down-regulated in DREADD+CNO group (n = 5) compared with that of control group (n = 5, median relative level in DG and CA3: 10.4 vs 8.4, 11.6 vs 7.8, respectively, Fig 5B).

Next, effects of 1, 5, and 10 mg/kg CNO on epileptogenesis, MFS and expressions of NDR2 were compared respectively by multiple comparisons. Kindling rate, mortality, latency and tendency of kindling, and severe seizure latency did not differ significantly between doses of CNO (Fig 6A–6E). Multiple comparisons of the seizure stage at all PTZ injection days were performed, in comparison to DREADD rats treated with 1 mg/kg CNO (n = 8), decreases in seizure score were observed in DREADD rats treated with 5 mg/kg CNO (n = 22) at day 8 and DREADD rats treated with 10 mg/kg CNO at day 8 and day 17 (Fig 6F).

Although no statistically significant difference in Timm score was observed in multiple comparisons, A trend toward lower Timm score in CA3 was observed in DREADD rats treated with CNO at 10 mg/kg(n = 6) as compared with DREADD rats treated with CNO at 1 mg/kg(n = 4, Fig 3C). Comparing NDR2 levels in CA3 and DG in DREADD rats treated with 1 mg/kg CNO (n = 4), decreased NDR2 in CA3 was observed in DREADD rats treated with 5 and 10 mg/kg CNO (n = 4 and 5, median relative level in CA3:9.9 vs 8.2 vs 7.8, respectively), and decreased NDR2 in DG was observed in DREADD rats treated with 10 mg/kg CNO (median relative level in DG: 10.4 vs 8.4,Fig 5C).

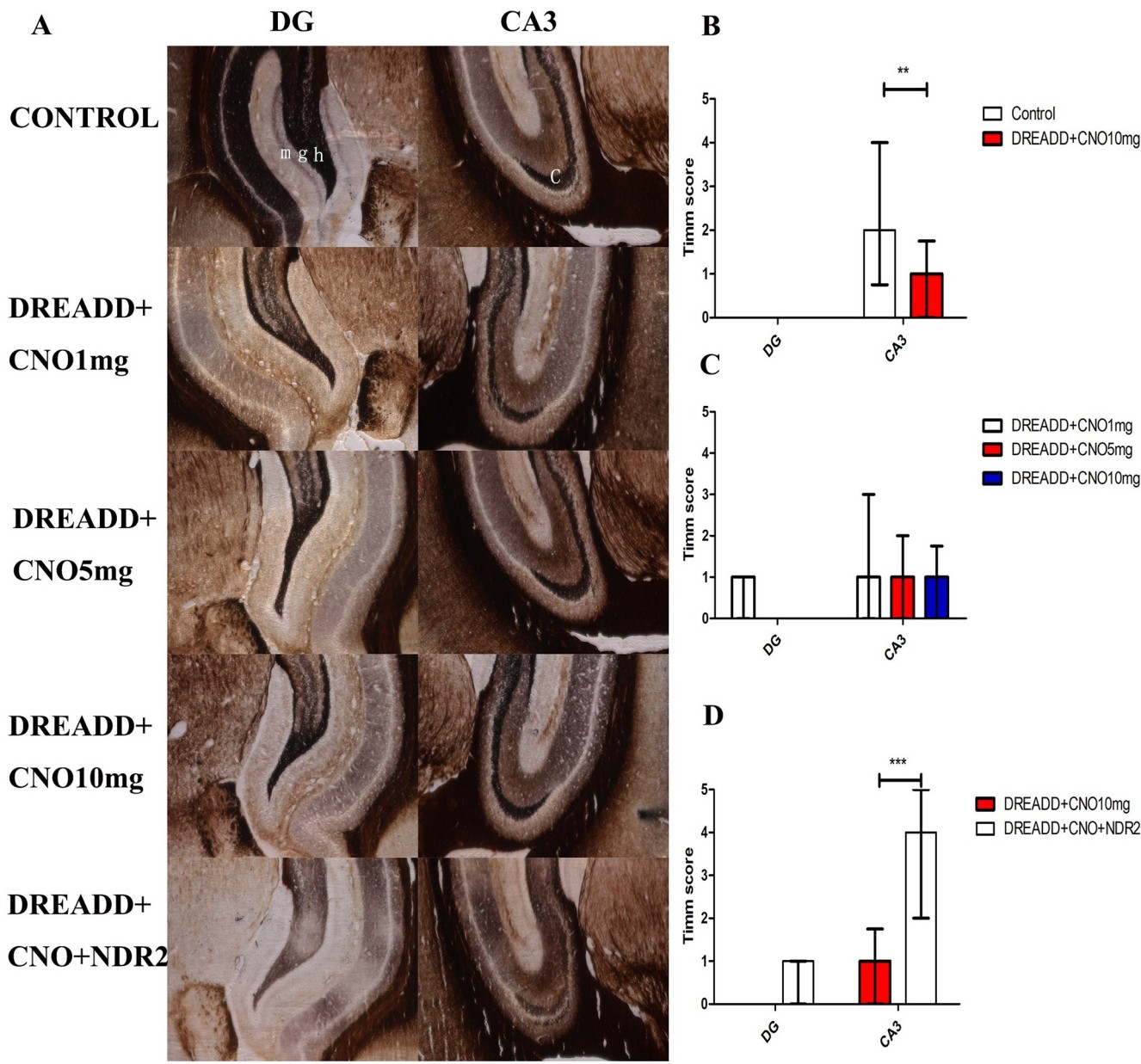

**Fig 3. MFS in control, DREADD+CNO and DREADD+CNO+NDR2 groups.** A, Representative pictures of MFS in the CA3 and DG regions; B, MFS of CA3 region in DREADD+CNO group(n = 6) was distinctly decreased in comparison with that in control group (n = 4, Mann-Whitney test); C, MFS of CA3 and DG were similar in DREADD rats treated with 1,5 and10 mg/kg CNO(n = 4,6 and 6 respectively, Kruskal-Wallis test); D, MFS of CA3 region in DREADD+CNO+NDR2 group(n = 5) was distinctly increased in comparison with that in DREADD+CNO group (Mann-Whitney test). CNO, clozapine N-oxide; DG, dentate gyrus; m, molecular layer; g, granule cell layer; h, hilus; c: CA3 region; All data represent the mean ± SD. *indicates $P < 0.05$, **indicates $P < 0.01$,*** indicates $P < 0.001$.

### 3.3 Overexpression of NDR2 in hippocampus

The expression of NDR2 was confirmed in the DREADD+CNO+NDR2 group. Data showed that the hippocampal level of NDR2 was significantly increased in DREADD+CNO+NDR2 group (n = 5) compared with that of DREADD+CNO group (median relative level:1.59vs 1.35, $p<0.01$, Fig 4B). In addition, immunofluorescence analysis demonstrated that the level of NDR2 in the DG and CA3 of the hippocampus was up-regulated in DREADD+CNO+NDR2

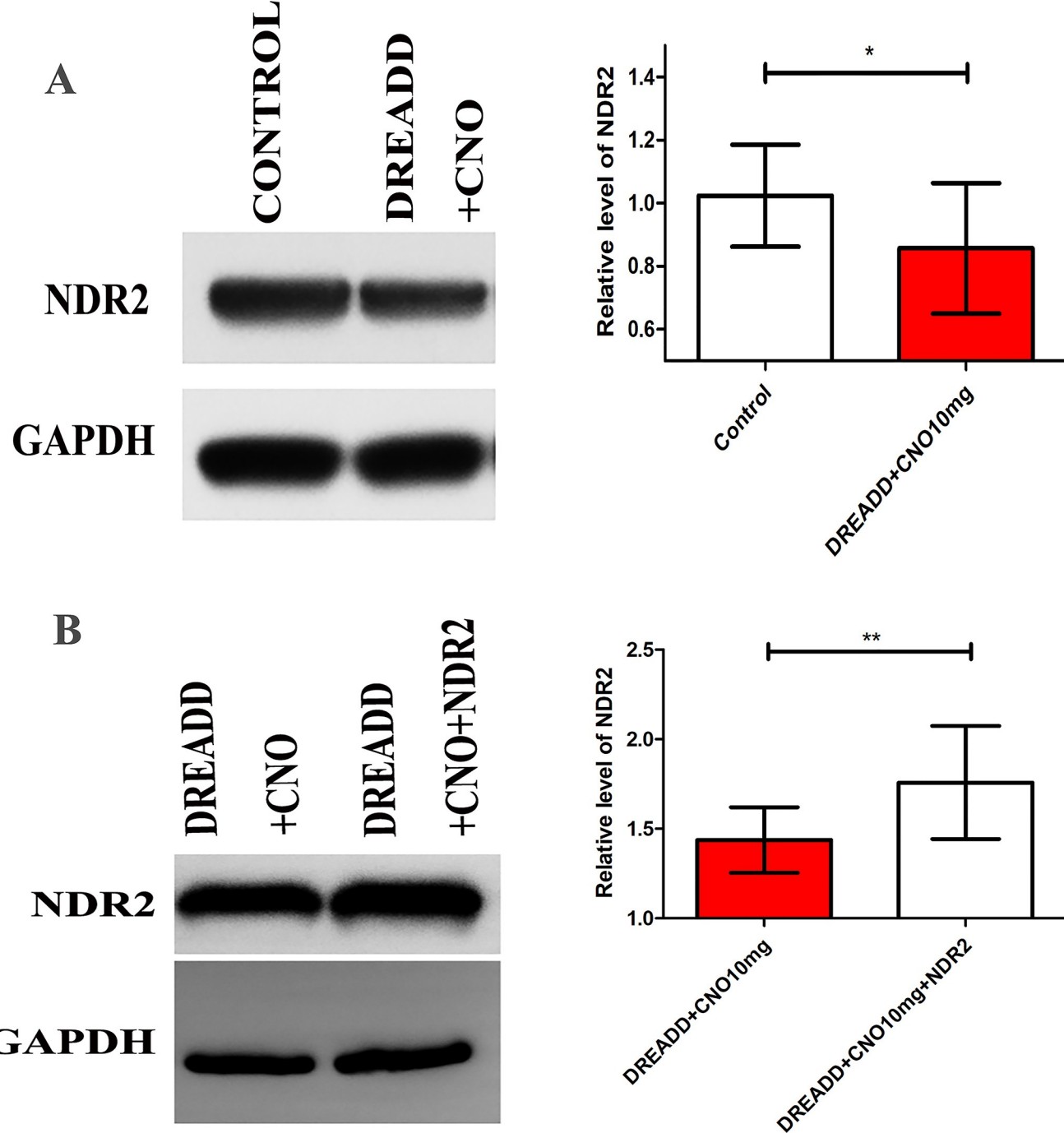

**Fig 4. NDR2 expression in control, DREADD+CNO and DREADD+CNO+NDR2 groups by Western blot analysis.** A, The level of NDR2 in DREADD +CNO group (n = 4) was decreased compared with that in control group (n = 4, Mann-Whitney test): B, The level of NDR2 in DREADD+CNO+NDR2 group (n = 4) was increased compared with that in DREADD+CNO group. CNO, clozapine N-oxide. All data represent the mean ± SD. *indicates $P < 0.05$, **indicates $P < 0.01$.

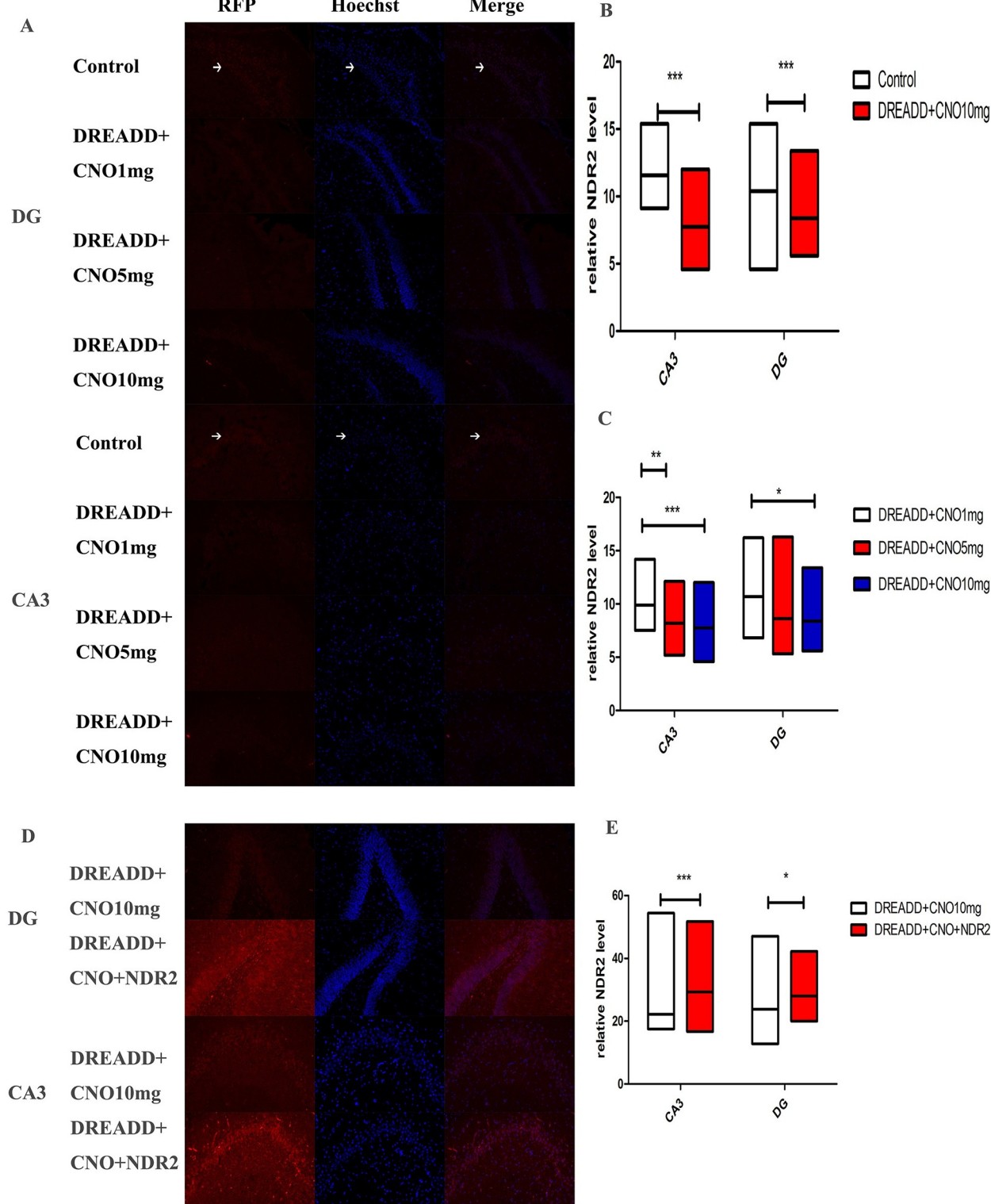

**Fig 5. The NDR2 levels in the control, DREADD+CNO and DREADD+CNO+NDR2 groups were measured by immunofluorescence.** A and D, Representative fluorescence pictures of the CA3 and DG are displayed; B, the level of NDR2 in the DG and CA3 of the hippocampus were down-regulated in DREADD+CNO group(n = 5) compared with that of control group(n = 5, Mann-Whitney test); C, Comparing NDR2 levels in CA3 and DG in DREADD rats treated with 1 mg/kg CNO(n = 4), decreased NDR2 in CA3 was observed in DREADD rats treated with 5 and 10 mg/kg CNO(n = 4 and 5 respectively), and decreased NDR2 in DG was observed in DREADD rats treated with 10 mg/kg CNO(Kruskal-Wallis test);E, The

level of NDR2 in the DG and CA3 of the hippocampus were up-regulated in DREADD+CNO+NDR2 group(n = 7) compared with that of DREADD+CNO group(Mann-Whitney test). CNO, clozapine N-oxide; DG, dentate gyrus. All data represent the median with range. *indicates $P < 0.05$, **indicates $P < 0.01$,*** indicates $P < 0.001$. White arrows show the positive cells.

group compared with that of DREADD+CNO group (n = 7, median relative level in DG and CA3:23.9 vs 28.0,22.2 vs 29.3, respectively, Fig 5E).

### 3.4 The suppressive effects of chemogenetic inhibition in DG on MFS and epileptogenesis were blocked by overexpression of NDR2 in the hippocampus

In comparison to DREADD+CNO group (treated with 10 mg/kg CNO), overexpression of NDR2 in hippocampus induced increase in mortality (from 8.3% to 32.3%) and kindling rate (from 63.6% to 95.2%), shorter latency of kindling (from 21.0±7.8 days to 13.4±5.6 days) and prominent alternation of kindling tendency (Fig 7A–7C and 7E) in the DREADD+CNO+NDR2 group(n = 21). Regarding seizure stage, we observed an obvious increase in seizure score in the DREADD+CNO+NDR2 group at day 3–8 and day 10-13(Fig 7F). No alteration of severe seizure latency was obtained by overexpression of NDR2 in the hippocampus (Fig 7D).

   We determined whether overexpression of NDR2 reversed MFS in DREADD rats. MFS of CA3 was significantly up-regulated in DREADD+CNO+NDR2 group (n = 5) compared with

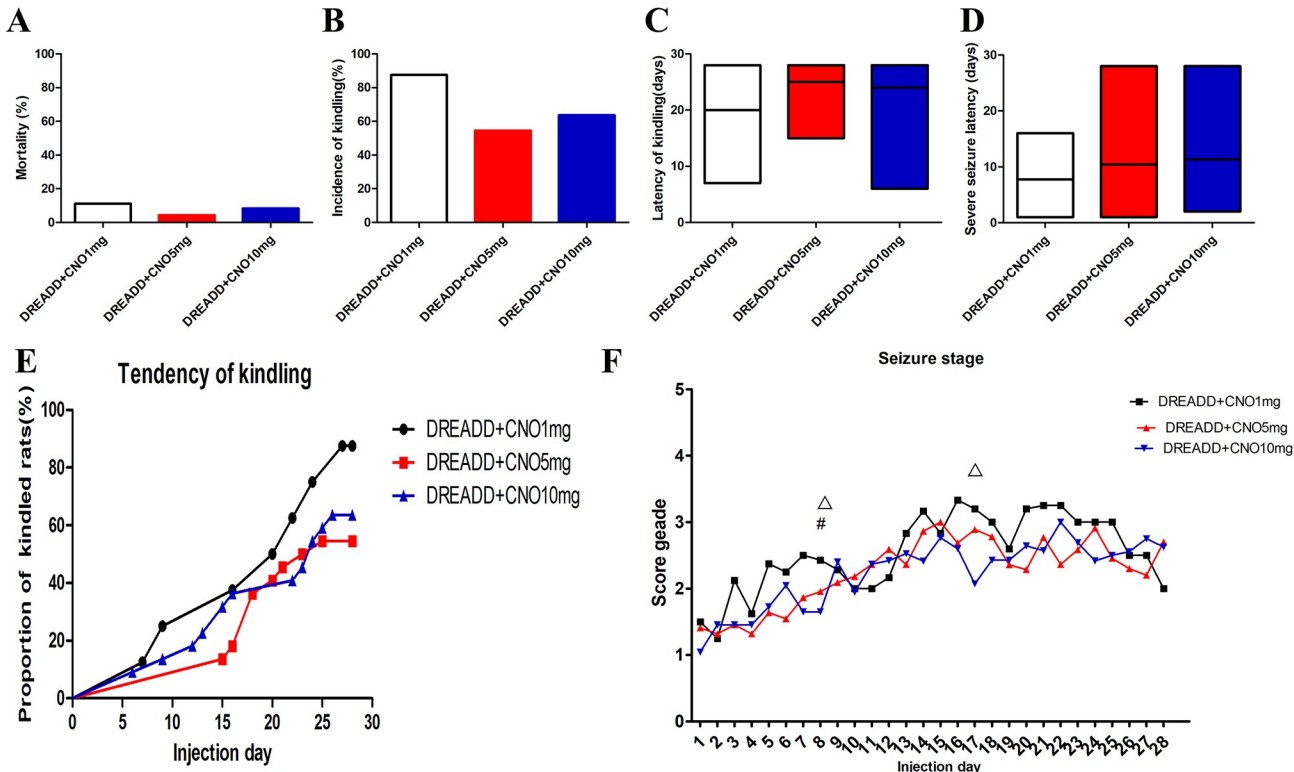

**Fig 6. The effect of multiple CNO doses on epileptogenesis in PTZ kindling rat model infected with hM4Di DREADD in DG.** Mortality(A), kindling rate(B), latency of kindling(C), severe seizure latency(D), tendency of kindling(E) and seizure stage(F) of DREADD rats with CNO at 1 mg/kg, 5mg/kg and 10 mg/kg. CNO, clozapine N-oxide; DG, dentate gyrus. Δindicates $P < 0.05$, 10mg/kg CNO vs 1mg/kg CNO; # indicates $P < 0.05$, 5mg/kg CNO vs 1mg/kg CNO.

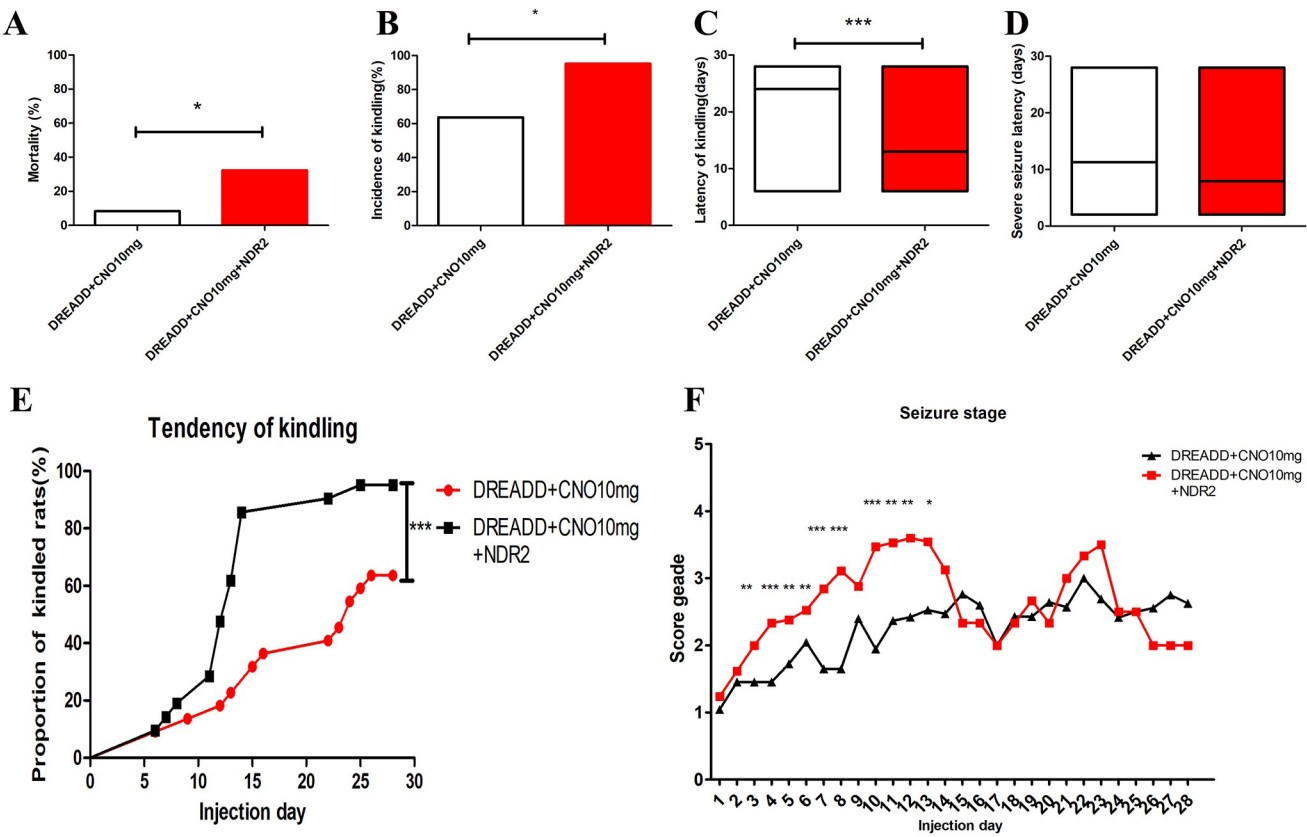

**Fig 7. NDR2 overexpression in hippocampus suppressesed the effect of CNO administration on epileptogenesis in PTZ kindling rat model infected with hM4Di vector in DG.** Mortality(A), kindling rate(B), latency of kindling(C), severe seizure latency(D), tendency of kindling(E) and seizure stage(F) between DREADD+CNO group and DREADD+CNO+NDR2 group. CNO, clozapine N-oxide; DG, dentate gyrus. *indicates $P < 0.05$, **indicates $P < 0.01$, ***indicates $P < 0.001$.

that of DREADD+CNO group (median Timm score:2.0 vs 4.0, $p<0.001$), and MFS in DG was not altered by expression of NDR2 in hippocampus (Fig 3D).

## 4. Discussion

The findings of our study revealed that chemogenetic silencing of the DG in the hippocampus prevent epileptogenesis in PTZ-kindling rat model of TLE, and such effects could be blocked by overexpression of NDR2 with development of MFS.

Epilepsy is a chronic neurological disorder characterized by recurrent epileptic seizures. DG plays a significant role in development of epilepsy as a potential therapeutic target for epilepsy. Dentate gate hypothesis argued that hippocampal circuits were protected by DG from overexcitation and that a breakdown of the dentate gate results in epilepsy. Krook-Magnuson et al found interventions selectively silencing the DG could inhibit seizures [12]. However, it is unclear whether silencing of DG would prevent epileptogenesis and the underlying mechanism deserves further exploration.

In the current study, we observed a significant suppression in seizure severity and kindling in DREADD rats with CNO administration at 10 mg/kg, which indicated chemogenetic silencing of DG in the hippocampus prevent epileptogenesis.

Previous studies had confirmed that transduction with hM4Di does not effect neuronal excitability in the absence of CNO [22–25]. In order to confirm that the effects observed in our

study were not due to a nonspecific action of CNO, we choose the highest dose of CNO (10 mg/kg) to test the possibility of non-specific CNO-induced effects in epileptogenesis, MFS and expression of specific protein. When treated with 10 mg/kg of CNO, no effect on seizures, kindling, MFS and protein expression was observed in rats injected with a control vector, which is consistent with the previous study [25, 26].

CNO is a specific agonist of DREADDs. The duration of effect is dictated by the half-life and doses of CNO. Although the half-life of CNO in rats has not been measured systematically, it is fully cleared within 12 h of administration of clozapine [27]. In addition, previous studies demonstrated that an anti-seizure effect that appears after acute CNO administration at high doses (10 mg/kg) only last 15 hours in DREADD-expressing experimental animals [28]. In the present study, CNO injection was conducted after thirty-minute seizure observation. Therefore, the Racine score during the behavioral observation period was influenced by consecutive daily CNO injections rather than a single administration of CNO 24 hours earlier.

Changes in axonal growth influence MFS. Previous studies found that because of hippocampal cell death granule cells in DG lose their target cells in CA4 and CA3 regions, which resulted in the formation of MFS [29], and extent of MFS correlates with neuronal cell loss in patients with epilepsy [30]. Although the presence of MFS implies the neuronal loss in the hippocampus, MFS is the best-studied form of axonal plasticity and reorganization in epilepsy, which has preceded the appearance of spontaneous seizures in the PTZ kindling rat model of epilepsy [31]. Previous studies found that MFS contributed to recurrent excitatory circuitry [32], and severity of MFS is associated with susceptibility of spontaneous seizures [33], which indicated that MFS contributed to the epileptogenesis of TLE. Previous studies had confirmed that axonal morphological plasticity was regulated by the excitability of neurons [13], and hyperactivity of granule cells resulted in the sprouting of mossy fiber in vitro [14]. In the present study, CNO administration suppressed MFS, seizure severity and kindling in DREADD-expressing rats, which means chemogenetic silencing of DG may prevent epileptogenesis via relieving MFS.

However, the molecular mechanism underlying such findings remains unknown. Many molecular mechanisms of MFS have been described, and some studies have indicated that proteins involved in regulating axonal outgrowth regulate the severity of MFS and seizures, such RGMa, PTEN and BDNF [10, 34, 35]. The mechanism by which modulating the regulator of axonal outgrowth suppresses seizures is unknown. but changes in neuronal hyperexcitability may contruibute to therapeutic effect. It has been demonstrated that MFS contributes to pathologic hyperexcitability of hippocampal circuits [36–38]. Hyperexcitability may be suppressed by alleviated MFS. For instance, Chen et al found overexpression of RGMa decreased NMDAR-mediated EPSC [10].

NDR2 is a protein kinase important for neuronal polarity and morphogenesis [39], and controls integrin-dependent neurite growth in mouse hippocampal neurons [40]. Moreover, the mossy fiber in the ventral hippocampus was regulated in transgenic mice overexpressing Ndr2 in the forebrain including the hippocampus and neocortex [41]. In the present study, relieved mossy fiber sprouting and decreased expression of NDR2 occurs in hM4Di-expressing rats with CNO administration and overexpression of NDR2 in hippocampus promote MFS and block protective effect of chemogenetic silencing of DG on epileptogenesis, which indicated chemogenetic silencing of DG suppress MFS and epileptogenesis through NDR2. Such a finding is similar to the previous report that overexpression of NDR2 resulted in increased growth of neurites in PC12 cells [42]. On the contrary, Madencioglu et al found mossy fiber was reduced in transgenic mice overexpressing Ndr2 in the forebrain [41]. Those findings indicated that NDR2 can be a bidirectional molecule regulating neurite outgrowth under different physiological and pathological conditions.

Substantia nigra is a mesencephalic structure inserted along several circuits which appear to play a key role in epilepsy [43]. The function of substantia nigra pars reticulata (SNpr) has been associated with the control of seizures. Behavioral seizures induced in the hippocampus are blocked by a GABAergic receptor agonist muscimol microinjected in the SNpr [44]. Moreover, the injection of exogenous DA into SNpr induced seizures and a significant reduction of gene expression for GluR1, GluR2 and NMDAR1 subunits in rat hippocampal subfields including CA1 through CA4 and DG [45], which indicated hippocampus-SNpr pathway may be involved in the mechanism of epilepsy. Whether the connection between hippocampus and SNpr is involved in the therapeutic effect of the modulator of axonal growth deserves further study.

## 5. Conclusions

Collectively, a deeper understanding of the effects of chemogenetic silencing of DG and Its modulator in epileptogenesis may help identify novel targets to arrest epileptogenesis and prevent epilepsy. Thus, the present study suggests that chemogenetic silencing of DG prevents epileptogenesis through NDR2 in the hippocampus.

## Supporting information

**S1 File. Raw data of statistical analysis.**
(ZIP)

**S1 Fig. Raw images of Western blotting.**
(PDF)

## Author Contributions

**Conceptualization:** Zheren Tan, Fafa Tian.

**Data curation:** Chen Zhang.

**Formal analysis:** Chen Zhang, Zixian He.

**Funding acquisition:** Fafa Tian.

**Investigation:** Chen Zhang.

**Methodology:** Chen Zhang.

**Project administration:** Fafa Tian.

**Supervision:** Zheren Tan, Fafa Tian.

**Validation:** Chen Zhang.

**Writing – original draft:** Chen Zhang.

**Writing – review & editing:** Zheren Tan, Fafa Tian.

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
