## [Decision Letter · Decision Letter 0]

29 Dec 2022

PONE-D-22-30742Silencing of dentate gyrus inhibits mossy fiber sprouting and prevents epileptogenesis through NDR2 kinase in pentylenetetrazole kindling rat model of TLEPLOS ONE

Dear Dr. Fafa Tian,

Thank you for submitting your manuscript to PLOS ONE. After careful consideration, we feel that it has merit but does not fully meet PLOS ONE’s publication criteria as it currently stands. Therefore, we invite you to submit a revised version of the manuscript that addresses the points raised during the review process.

We look forward to receiving your revised manuscript.

Kind regards,

Andreia Cristina Karklin Mortensen, Ph.D.

Academic Editor

PLOS ONE

Journal Requirements:

3. Please amend your manuscript to include your abstract after the title page.

Additional Editor Comments:

The manuscript “Silencing of dentate gyrus inhibits mossy fiber sprouting and prevents epileptogenesis through NDR2 kinase in pentylenetetrazole kindling rat model of TLE”, by Zhang et al establishes mechanistic studies the role of mossy fiber sprouting on epileptogenesis using chemogenetic approaches. The authors conclude that silencing of the dentate gyrus inhibits sprouting and prevents epileptogenesis through NDR2, suggesting a potential therapeutic strategy. To be amenable for publication, please address the criticisms and suggestions by reviewer 1. Additionally, the English language must be thoroughly revised by a native speaker, as several sentences and paragraphs are not comprehensible.

Reviewers' comments:

Reviewer's Responses to Questions

**Comments to the Author**

1. Is the manuscript technically sound, and do the data support the conclusions?

Reviewer #1: Yes

2. Has the statistical analysis been performed appropriately and rigorously? 

Reviewer #1: Yes

3. Have the authors made all data underlying the findings in their manuscript fully available?

Reviewer #1: Yes

4. Is the manuscript presented in an intelligible fashion and written in standard English?

Reviewer #1: Yes

5. Review Comments to the Author

Reviewer #1: The manuscript titled “Silencing of dentate gyrus inhibits mossy fiber sprouting and prevents epileptogenesis through NDR2 kinase in pentylenetetrazole kindling rat model of TLE”, by Fafa Tian and colleagues demonstrates an experimental epileptogenesis study. The authors point out that the role of DG’s excitability in epileptogenesis have not yet been well investigated, and underlying mechanism have not been elucidated.

After careful reading, I suggest the following to improve the current manuscript:

- In the following text: “…The commonest type of refractory epilepsy that can be found in adults is the temporal lobe epilepsy (TLE)[3]”… Are these adults’ humans or experimental animals? Please detail this type of information.

- What country is the “National Institutes of Health for the care and use of laboratory animals” from? The authors already cited the animal experimental rules and protocol number.

- Where is Central South University? Please add in the text.

- There are many mentions of animal ethics committees, such as: “Ethics Committee of Xiangya Hospital, Central South University (Protocol Number: 20153213”); “guidelines of the World Medical Association Declaration of Helsinki”; “Department of experimental animals, Central South University”; “ARRIVE (Animal Research: Reporting of In Vivo Experiments) guidelines”. Please explain all these committees and guidelines.

- How many animals per group (I, II and III) were used?

- In the following text: ”…and the others were euthanized after the last observation of PTZ-induced seizure...”. After all, how many animals were euthanized?

- What was the stereotaxic device used (company, brand, model, etc.)?

- There are many errors in English grammar, concordance, typing, etc. (examples: "sterotaxic" device), please correct. Check and standardize in the manuscript. I strongly recommend that you ask the assistance of a native English speaker to revise the whole manuscript.

- Where is "OBiO Technology" from? Where is MultiScience from? Please check and standardize the origin of the equipment’s and drugs in the manuscript.

- How much rats died during the course of PTZ kindling?

- What equipment is used for coronal sections frozen cryosections?

- Photomicrographs were captured for each animal with a microscope. What was this microscope?

- What are Beyotime, Servicebio, etc?

- In Figure 1, it would be better to indicate some structure, cell, etc.

- In figures 3 and 5 indicate with symbols what we are seeing. In figure 3 the images are small. It would be good if the authors put an excerpt for each image with higher resolution.

- The findings indicated that chemogenetic inhibition in DG alleviates epileptogenesis in PTZ kindling rat model. How much were the values in, for example, percentage?

- The MFS and NDR2 expression in hippocampus were investigated. Is it more coherent to express with numbers, percentage, not only was it significant, how much better or worse, if it increased or decreased significantly how much was it?

- One more example: The expression of NDR2 was confirmed in DREADD+CNO+NDR2 group. Data showed that the hippocampal level of NDR2 was significantly increased in DREADD+CNO+NDR2. How much?

- In the last item (3.4) of the "results" the authors put the values in percentage! Therefore, it is also possible to place the results in this way for the other results, in addition to indicating the statistic as they did.

- Please discuss in the “Discussion”: Some authors consider that “Mossy fiber sprouting (MFS”) is a cell death marker phenomenon. Others, however, consider that the neuron is recovering, including a form of neuronal plasticity, regenerating. How do the authors see this aspect?

- Put in the “Discussion”: The “...regulators of axonal growth may be a potential target to prevent and cure TLE.” How specifically can this occur?

- A counterpoint to the previous consideration can be the study by Rodrigues et al. 2002 (doi: 10.1016/j.eplepsyres.2004.02.001): "...present results are the first behavioral description—and comparison—of seizures induced by ICV and subregional hippocampal formation (dorsal × ventral) bicuculline microinjection in rats. Also, this is the first attempt to block the seizures originated by focal bicuculline in the dorsal hippocampus with GABAergic drugs microinjected in the “substantia nigra pars reticulata “ (SNPR) of rats...". How do the authors see this manipulation (dorsal x ventral) and the one they propose to resolve, in part, temporal lobe epileptic (TLE) seizures? Is there any link between the two ideas (hippocampus-DG….SNPR)? Therefore, due to such a connection with another area close to the brain.

- Still in this direction, regulators of axonal growth may be a potential target to prevent and cure TLE are they efficient due to connections (pathways- axons) linking the hippocampus to the SNPR. Please discuss.

6. PLOS authors have the option to publish the peer review history of their article (what does this mean?). If published, this will include your full peer review and any attached files.

Reviewer #1: **Yes: **Wagner Ferreira dos Santos

---

## [Author Response · Author response to Decision Letter 0]

16 Jan 2023

Reviewer #1: The manuscript titled “Silencing of dentate gyrus inhibits mossy fiber sprouting and prevents epileptogenesis through NDR2 kinase in pentylenetetrazole kindling rat model of TLE”, by Fafa Tian and colleagues demonstrates an experimental epileptogenesis study. The authors point out that the role of DG’s excitability in epileptogenesis have not yet been well investigated, and underlying mechanism have not been elucidated.

After careful reading, I suggest the following to improve the current manuscript:

- In the following text: “…The commonest type of refractory epilepsy that can be found in adults is the temporal lobe epilepsy (TLE)[3]”… Are these adults’ humans or experimental animals? Please detail this type of information.

Thanks for your suggestion. Those adults is human.The sentence has been rewritten. 

- What country is the “National Institutes of Health for the care and use of laboratory animals” from? The authors already cited the animal experimental rules and protocol number.

Thanks for your question. “National Institutes of Health is NIH of USA. Although this study was conducted in China, but the animal research ethics standards of United States is worthy of our reference.

- Where is Central South University? Please add in the text.

CSU is located in Changsha, China. Corresponding content has been added in manuscript.

- There are many mentions of animal ethics committees, such as: “Ethics Committee of Xiangya Hospital, Central South University (Protocol Number: 20153213”); “guidelines of the World Medical Association Declaration of Helsinki”; “Department of experimental animals, Central South University”; “ARRIVE (Animal Research: Reporting of In Vivo Experiments) guidelines”. Please explain all these committees and guidelines.

Thanks for your question. This study was conducted in China.As we all known, Western countries have more excellent practical experience and theory in animal protection, so the animal research and care standards in this study refer to the famous guidelines you mentioned, and finally the project was approved by Ethics Committee of Xiangya Hospital.The experiments were conducted in Department of experimental animals, Central South University.

- How many animals per group (I, II and III) were used?

Thanks for your question,12 rats were used in group I , 11 rats were used in group II, 56rats were used in group III,30 rats were used in group IV.

- In the following text: ”…and the others were euthanized after the last observation of PTZ-induced seizure...”. After all, how many animals were euthanized?

Thanks for your question.96 rats were euthanized.

- What was the stereotaxic device used (company, brand, model, etc.)?

Thanks for your question. The stereotaxic device was purchased from Yuyan Instruments, China.The brand information was added.

- There are many errors in English grammar, concordance, typing, etc. (examples: "sterotaxic" device), please correct. Check and standardize in the manuscript. I strongly recommend that you ask the assistance of a native English speaker to revise the whole manuscript.

I am so sorry for my poor English.The manuscript has been polished.

- Where is "OBiO Technology" from? Where is MultiScience from? Please check and standardize the origin of the equipment’s and drugs in the manuscript.

Thanks for your question and suggestion.Those companies is from China.The origin of drugs and equipments has been added in methods section.

- How much rats died during the course of PTZ kindling?

Thanks for your question.13 rats died. Most of them is from group IV.

- What equipment is used for coronal sections frozen cryosections?

Leica CM1950.

- Photomicrographs were captured for each animal with a microscope. What was this microscope?

Thanks for your question. The Nikon eclipse C1 was used.

- What are Beyotime, Servicebio, etc?

Those are Chinese technology companies.

- In Figure 1, it would be better to indicate some structure, cell, etc.

Thanks for your suggestion, the layers were indicated by smybols in revised Figure 1.

- In figures 3 and 5 indicate with symbols what we are seeing. In figure 3 the images are small. It would be good if the authors put an excerpt for each image with higher resolution.

Thanks for your suggestion, symbols were added to figure 3and 5,the high resolution figure 3 was uploaded.

- The findings indicated that chemogenetic inhibition in DG alleviates epileptogenesis in PTZ kindling rat model. How much were the values in, for example, percentage?

The kindling rate was decreased from 91.7%(group I) to 63.6(group III with CNO 10mg/kg)，the mean latency of kindling was prolonged from 15.4days（group I）to 21.0（group III with CNO 10mg/kg）.

- The MFS and NDR2 expression in hippocampus were investigated. Is it more coherent to express with numbers, percentage, not only was it significant, how much better or worse, if it increased or decreased significantly how much was it?

Thanks for your suggestion,The median timm score and relative level of protein had been added in result section to show the difference between groups.

- One more example: The expression of NDR2 was confirmed in DREADD+CNO+NDR2 group. Data showed that the hippocampal level of NDR2 was significantly increased in DREADD+CNO+NDR2. How much?

Thanks for your suggestion,The median timm score and relative level of protein had been added in result section to show the difference between groups.

- In the last item (3.4) of the "results" the authors put the values in percentage! Therefore, it is also possible to place the results in this way for the other results, in addition to indicating the statistic as they did.

Thanks for your suggestion. Findings were presented with such way in other part of result section.

- Please discuss in the “Discussion”: Some authors consider that “Mossy fiber sprouting (MFS”) is a cell death marker phenomenon. Others, however, consider that the neuron is recovering, including a form of neuronal plasticity, regenerating. How do the authors see this aspect?

Thanks for your suggestion. Previous studies found that because of hippocampal cell death granule cells in DG lose their target cells in CA4 and CA3 regions, which resulted in the formation of MFS ,and extent of MFS correlates with neuronal cell loss in patients with epilepsy.Although presence of MFS implies the neuronal loss in hippocampus, MFS is the best-studied form of axonal plasticity and reorganization in epilepsy, which contributed to development of epilepsy.

- Put in the “Discussion”: The “...regulators of axonal growth may be a potential target to prevent and cure TLE.” How specifically can this occur?

Thanks for your suggestion, relative contents has been added in discussion section.

- A counterpoint to the previous consideration can be the study by Rodrigues et al. 2002 (doi: 10.1016/j.eplepsyres.2004.02.001): "...present results are the first behavioral description—and comparison—of seizures induced by ICV and subregional hippocampal formation (dorsal × ventral) bicuculline microinjection in rats. Also, this is the first attempt to block the seizures originated by focal bicuculline in the dorsal hippocampus with GABAergic drugs microinjected in the “substantia nigra pars reticulata “ (SNPR) of rats...". How do the authors see this manipulation (dorsal x ventral) and the one they propose to resolve, in part, temporal lobe epileptic (TLE) seizures? Is there any link between the two ideas (hippocampus-DG….SNPR)? Therefore, due to such a connection with another area close to the brain.

From the perspective of seizure model creation, IVC injection can induce seizures faster and have more diverse seizure manifestations. However, due to the wide range affected by IVC injections, the relationship between specific brain regions and epileptic manifestations cannot be established. So I think it is brilliant to conduct the local injection in DHF and AHiPM, and make a comparison in seizure behaviors between IVC injection and local injection,which can better help us understand the origins of seizure behaviors heterogeneity.

- Still in this direction, regulators of axonal growth may be a potential target to prevent and cure TLE are they efficient due to connections (pathways- axons) linking the hippocampus to the SNPR. Please discuss.

Thanks for your ssuggestion. Substantia nigra is a mesencephalic structure inserted along several circuits which appear to play a key role in epilepsy. The function of substantia nigra pars reticulata (SNpr) has been associated with the control of seizures. Behavioral seizures induced in the hippocampus are blocked by a GABAergic receptor agonist muscimol microinjected in the SNpr. Moreover, injection of exogenous DA into SNpr induced seizures and a significant reduction of gene expression for GluR1, GluR2 and NMDAR1 subunits in rat hippocampal subfields including CA1 through CA4 and DG,which indicated hippocampus-SNpr pathway may be involved in mechanism of epilepsy. Whether the connection between hippocampus and SNpr is involved in therapeutic effect of modulator of axonal growth deserves further study.

---

## [Editor Report · Decision Letter 1]

29 Mar 2023

Silencing of dentate gyrus inhibits mossy fiber sprouting and prevents epileptogenesis through NDR2 kinase in pentylenetetrazole kindling rat model of TLE

PONE-D-22-30742R1

Dear Dr.  Fafa Tian

We’re pleased to inform you that your manuscript has been judged scientifically suitable for publication and will be formally accepted for publication once it meets all outstanding technical requirements.

Kind regards,

Andreia Cristina Karklin Mortensen, Ph.D.

Academic Editor

PLOS ONE
---

## [Editor Report · Acceptance letter]

3 Apr 2023

PONE-D-22-30742R1 

Silencing of dentate gyrus inhibits mossy fiber sprouting and prevents epileptogenesis through NDR2 kinase in pentylenetetrazole kindling rat model of TLE 

Dear Dr. Tian:

I'm pleased to inform you that your manuscript has been deemed suitable for publication in PLOS ONE. Congratulations! Your manuscript is now with our production department. 

Kind regards, 

on behalf of

Dr. Andreia Cristina Karklin Mortensen 

Academic Editor

PLOS ONE